# Impact of COVID-19 Pandemic on Delay of Melanoma Diagnosis: A Systematic Review and Meta-Analysis

**DOI:** 10.3390/cancers16223734

**Published:** 2024-11-05

**Authors:** Cristina Pellegrini, Saverio Caini, Aurora Gaeta, Eleonora Lucantonio, Mirco Mastrangelo, Manfredo Bruni, Maria Esposito, Chiara Doccioli, Paola Queirolo, Giulio Tosti, Sara Raimondi, Sara Gandini, Maria Concetta Fargnoli

**Affiliations:** 1Department of Biotechnological and Applied Clinical Sciences, University of L’Aquila, 67100 L’Aquila, Italy; cristina.pellegrini@univaq.it (C.P.); eleonora.lucantonio@graduate.univaq.it (E.L.); mirco.mastrangelo@univaq.it (M.M.); manfredo.bruni@graduate.univaq.it (M.B.); maria.esposito3@univaq.it (M.E.); 2Cancer Risk Factors and Lifestyle Epidemiology Unit, Institute for Cancer Research, Prevention and Clinical Network (ISPRO), 50139 Florence, Italy; s.caini@ispro.toscana.it; 3Department of Statistics and Quantitative Methods, University of Milano-Bicocca, 20125 Milan, Italy; aurora.gaeta@ieo.it; 4Department of Experimental Oncology, European Institute of Oncology, 20139 Milan, Italy; sara_raimondi@hotmail.com; 5UOSD General and Oncologic Dermatology, San Salvatore Hospital, 67100 L’Aquila, Italy; 6Clinical Epidemiology Unit, Institute for Cancer Research, Prevention and Clinical Network (ISPRO), 50139 Florence, Italy; c.doccioli@ispro.toscana.it; 7Division of Melanoma Sarcoma and Rare Tumors, IRCCS European Institute of Oncology, 20141 Milan, Italy; paola.queirolo@ieo.it; 8Dermato-Oncology Unit, European Institute of Oncology, 20141 Milan, Italy; giulio.tosti@ieo.it

**Keywords:** COVID-19, melanoma, diagnostic delay, prognosis

## Abstract

Previous studies from sparse experience demonstrated how restrictive measures due to COVID-19 resulted in delay of melanoma diagnoses leading to an advanced disease with a more severe prognosis. Following our systematic review and meta-analysis, which summarize all published data, a significantly higher prevalence of worse prognostic factors, such as depth thickness; ulceration, mitosis, and advanced stages, were found in melanoma diagnosed during the post-lockdown period compared to the pre-COVID-19 phase. The disruption of healthcare systems caused by the COVID-19 pandemic resulted in a significant shift in melanoma diagnoses towards more advanced lesions with a less favorable prognosis. This emphasizes the importance of creating and updating pandemic preparedness plans to limit the impact of any future events on oncological care.

## 1. Introduction

Cutaneous melanoma is an aggressive skin malignancy responsible for nearly 60,000 deaths globally each year [1]. The incidence is 10–25 new cases per 100,000 in Europe and 30–60 cases per 100,000 in North America and Australia. The incidence is increasing globally, especially in male patients over 60 years of age and among populations with fair skin [2]. Given its high potential to metastasize, early melanoma diagnosis is crucial to prevent progression [3]. Diagnostic delays result in a worsening of all prognostic factors, including Breslow thickness, ulceration, mitotic rate, and metastasis extent, leading to reduced disease-specific survival [4].

In March 2020, the WHO declared a global pandemic due to the spread of a new disease caused by the COVID-19 coronavirus [5]. In response to the pandemic, many countries worldwide implemented a lockdown period to reduce the spread of the virus and alleviate the burden on the healthcare system. This measure aimed to reduce the number of infections and limit the potential for healthcare facilities to become overwhelmed by the outbreak [6]. Throughout the pandemic, dermatological services, including all skin cancer screenings, experienced a decline of more than 75% [7].

Various reports, mainly based on single-center experiences, have highlighted how anti-COVID-19 measures delayed melanoma diagnoses [8,9,10,11,12,13,14,15,16,17,18,19,20,21,22,23,24,25,26,27,28,29,30,31,32,33,34,35,36,37,38,39,40,41,42,43,44,45,46,47,48,49,50,51,52,53,54,55]. Consequently, the percentage of cases identified in more advanced stages has increased. A cumulative analysis that summarizes the sparse findings in this context could help to understand the real impact of the COVID-19 pandemic on melanoma. Two meta-analyses were previously published in this context [56,57]. The first, by Seretis et al. (2022), focused on three prognostic aspects as thickness, ulceration and stage, considering only studies from European countries published until 2022 [56]. The second, by Pablo Díaz-Calvillo and colleagues (2024), reviewed the main prognostic factors but restricted the meta-analysis only to Breslow thickness and ulceration [57].

Melanoma is a rapidly progressing malignancy with a high potential for metastasis, and the stage at which it is diagnosed is critical in determining the patient’s prognosis. Delays in melanoma diagnosis, such as those observed during the COVID-19 pandemic due to the suspension of routine dermatological services, may lead to the worsening of key prognostic factors and a higher percentage of melanomas being detected at more advanced stages. This might have a direct impact on disease-specific survival rates and increase mortality. Increased Breslow thickness, ulceration, and the presence of metastases have all been shown to correlate with lower survival probabilities and worse overall patient outcomes [58]. An important study highlighted that, in the UK, delays in the diagnosis of various cancers due to the COVID-19 pandemic could lead to a significant increase in mortality over the next 5 years. A further study found that a 3-month delay in cancer diagnosis led to a significant reduction in OS, with estimates suggesting a reduction of 4.8–16.6% in 5-year OS depending on the cancer type. The authors also highlighted the potential impact of delays in treatment on PFS, as advanced-stage disease would progress faster and require more aggressive, less effective treatment [59].

Despite not all studies directly reporting long-term OS due to the relatively short post-pandemic follow-up periods, both modeling and retrospective observational studies indicate that the pandemic has likely shortened both PFS and OS across multiple cancer types. The cumulative effect of reduced screening, delayed diagnoses, and interrupted treatment regimens has resulted in worsened cancer prognoses around the globe.

Our meta-analysis examined the potential delay in the diagnosis of melanoma during the pandemic and its impact on clinical and prognostic factors. For a more comprehensive view of the phenomenon, we analyzed data on all histopathological features related to the prognosis from a large worldwide sample of patients.

## 2. Materials and Methods

### 2.1. Study Design

A literature search in MEDLINE, EMBASE, and Scopus was conducted in September 2023. The string search “Cutaneous Melanoma” AND “COVID-19” was applied. No time, geographical, or language restrictions were applied (as long as an English abstract was available to define eligibility). A manual search of the reference lists of potentially eligible studies was also performed.

We included studies reporting data on new diagnoses of cutaneous melanoma in adult patients during and/or after lockdown compared to those diagnosed before the COVID-19 pandemic. We included only studies published in peer-reviewed journals that contained information on the histopathological characteristics of melanomas. The endpoints considered in publications were Breslow thickness, ulceration, histopathological subtype, presence of mitosis and AJCC stage. We excluded reports of therapeutic regimens for melanoma, editorials, and conference papers. After removing duplicates, papers were screened based on their title and abstract by two reviewers, and the full text of those potentially eligible for inclusion was read. Whenever there was a disagreement on a paper, this was passed to the next step of the selection process to minimize the risk that a relevant paper was incorrectly discarded.

### 2.2. Data Collection and Meta-Analysis

Data on study design, patients, and melanoma characteristics were extracted from eligible papers and entered into an internally standardized spreadsheet. In detail, we retrieved the following information: country and year in which the study was conducted, study size, number of patients, age, sex, melanoma site, histological subtype, Breslow thickness, number of mitoses, and stage at diagnosis.

Our primary outcome was the comparison of melanoma thickness between the pre-COVID-19 and post-lockdown periods. The secondary outcomes were the evaluation of the histopathological subtype, stage, and the presence of ulceration and mitosis in melanomas diagnosed in these two pandemic phases.

The clinico-pathological variables were grouped as follows: Breslow thickness (<1 mm, 1–2 mm, >2 mm; and in situ vs. invasive); ulceration (presence, absence); stage at diagnosis following the AJCC 8th edition (I, II, III, IV); histopathological subtype (superficial spreading melanoma SSM, nodular melanoma NM, acral melanoma ALM, lentigo maligna melanoma LMM); mitosis (presence, absence).

For the analysis, we categorized the data into two temporal groups: pre-COVID-19 (patients diagnosed before the first lockdown, set in March 2020) and post-lockdown (patients diagnosed from March 2020 to March 2022), and performed a dichotomous comparisons as follows: each Breslow category was compared to <1 mm; presence of ulceration vs. absence; each stage group was compared to stage I; each melanoma subtype was compared to SSM; presence of mitosis vs. absence.

The meta-analysis was conducted using a predetermined protocol established according to the Cochrane Handbook’s recommendations [60]. Due to the nature of the exposure being studied, all included studies were observational in design. As a result, the quality and potential biases of the original articles were evaluated using the Newcastle–Ottawa Scale (NOS), with separate assessments for studies based on their prospective or retrospective design (i.e., cohort and nested case–control studies for prospective, and non-nested case–control studies for retrospective), as well as for cross-sectional studies. Study quality was assessed according to three domains (selection, comparability, and exposure or outcome) using the NOS tool. The maximum score is nine, and a score of seven or above generally indicates good quality, meaning a low risk of bias.

In Appendix A, the NOS scale is reported, as well as the points accumulated for each section, namely ‘Selection’, ‘Comparability’, and ‘Exposure’. A maximum of four points can be awarded for the ‘Selection’ and ‘Exposure’ section, and a maximum of two for ‘Comparability’.

The quality of the review adhered to the updated PRISMA (Preferred Reporting Items for Systematic Reviews and Meta-Analyses) guidelines [61]. We calculated the estimate of Odds Ratios (ORs) and 95% Confidence Interval (CI) from crude data or used the adjusted ORs and corresponding 95% CI when available. Study-specific ORs were transformed into logOR and we used Woolf’s formula to evaluate the standard error of the logOR and calculate the corresponding 95% CI. A meta-analysis was carried out following the random effects model with Restricted Maximum Likelihood Estimation to pool the study-specific logOR into a summary odds ratio (SOR). The between-study heterogeneity was assessed via Higgins’s I^2^ statistics [62]. I^2^ was considered high when greater than 50%. Forest plots were generated to present the effect sizes of each study, accompanied by the 95% CIs. When heterogeneity among studies was present, a sensitivity analysis and meta-regression were conducted to investigate the potential effect of publication year, geographical location, study quality, and adjustments in the estimates on the observed heterogeneity. Exploratory analyses were performed using trim and/or fill analysis to investigate and adjust the SOR estimate when publication bias was suggested. Publication bias was assessed with the Begg and Egger test. The meta-analysis was conducted using R version 4.0.0. All reported *p*-values were two-sided and a *p*-value < 0.05 was considered statistically significant.

## 3. Results

### 3.1. Study Selection and Data Extraction

The literature search identified 588 non-duplicate papers, of which 524 were excluded based on their title and/or abstract. Sixty-five papers were read in full copy, and eight did not meet the eligibility criteria. From the 57 eligible studies, only 45, reported or allowed to estimate independent ORs for at least one of the established outcomes, and were included [8,9,10,11,12,13,14,15,16,17,18,19,20,21,22,23,24,25,26,27,28,29,30,31,32,33,34,35,36,37,38,39,40,41,42,43,44,45,46,47,48,49,50,51,52]. The selection process for the papers is reported in Figure 1.

Among the included studies, the majority were conducted in Italy (11/45, 24.4%%), followed by the USA (7/45, 15.5%), and Ireland (4/44, 8.8%). The papers were all published between 2020 and 2023. All studies were observational, with most of them (23/45, 51.1%) published as research letters, while 16 (16/45, 35.5%) were published as full-length articles, and the remainder were published as short reports or case reports (6/45, 13,3%). The studies’ characteristics are reported in Table 1.

There was considerable variability across the studies in terms of the clinical factors used, either independently or in combination, to estimate the diagnostic delay. In detail,. the most frequent outcome was the Breslow thickness, considered in 29 of 45 (64.4%) papers, followed by ulceration, in 24 (24/4, 53.3%), histopathological subtype, in 15 (15/45, 33.3%), tumor stage, in 12 (12/45, 26.6%), and mitosis, in 6 (6/44, 13.3%) (Table 1).

In Appendix A, the NOS scale is reported, as well as the points accumulated for each section. Fifteen studies were awarded four points, while thirty studies were awarded less than four. The PRISMA checklist is shown in Appendix A.

### 3.2. Prognostic Factors

#### 3.2.1. Thickness

From 26 studies, accounting for 83,542 patients (43,918 patients pre-COVID-19 and 39,624 post-lockdown), we found a significantly higher proportion of invasive melanomas than in situ melanomas in the post-lockdown period (SOR = 1.31, 95%CI 1.14–1.50; I^2^ = 74.45%, Begg *p* = 0.38, Egger *p* = 0.47) (Figure 2A).

By removing the potential outlier study of Ruggiero et al. (2023), we observed a slight decrease in the SOR and a reduction in the heterogeneity (SOR = 1.23, 95%CI 1.11–1.38, I^2^ = 54.30%, Begg *p* = 0.32, Egger *p* = 0.14). The meta-regression did not show any effect of the publication year (*p*-value = 0.74), geographic location (EU vs. not EU, *p* = 0.49), or the study quality (*p* = 0.22) on the final estimate. Moreover, we found a significant increase in thick melanomas in the post-lockdown group, considering both tumors with Breslow thickness 1–2 mm (Figure 3A, SOR = 1.14, 95%CI 1.08–1.20, I^2^ = 0%, Begg *p* = 1.0; Egger *p* = 0.95) and >2 mm (Figure 3B, SOR = 1.62, 95%CI 1.08–2.40, I^2^ = 56.3%, Begg *p* = 0.48; Egger *p* = 0.18) vs. melanomas < 1 mm. The heterogeneity between the studies in Figure 3B is mainly due to the effect of Balakirski et al. (2022) [24], whose exclusion returned an SOR of 1.90 (95% CI 1.39–2.59), with zero heterogeneity (I^2^ = 0%).

#### 3.2.2. Ulceration

Overall, 24 papers analyzed melanoma ulceration, including a total of 117,231 patients, 73,118 in the pre-COVID-19 group and 44,113 in the post-lockdown group. The rate of ulcerated tumors was higher in the post-lockdown period (SOR = 1.35, 95% CI 1.18–1.54) (Figure 2B), although the between-study heterogeneity was high (I^2^ = 78.39%). The observed heterogeneity did not depend on the year of publication (*p* = 0.28 for 2021 papers; *p* = 0.10 for 2022; *p* = 0.11 for 2023, with 2020 as reference), geographic location (EU vs. not EU, *p* = 0.54), use of adjustments in the estimates (not adjusted vs. adjusted, *p* = 0.39), or the study quality (*p* = 0.63). The Funnel plot showed the presence of a slight grade of asymmetry, confirmed by Egger’s test (*p* = 0.014). The SOR adjusted for publication bias remained significantly higher in the post-lockdown group compared to the pre-COVID-19 group (SOR = 1.25, 95% 1.10–1.42).

#### 3.2.3. AJCC Stage

For the analysis of tumor stage, we considered 18 papers, including 127,154 patients (78,495 in the pre-COVID-19 group and 48,659 in the post-lockdown group). We found a significantly higher rate of stage II patients versus stage I in the post-lockdown phase (SOR = 1.15, 95%CI 1.05–1.27; I^2^ = 11.9%) (Figure 4A), with no publication bias. Considering the advanced disease, both stage III and IV patients were more frequent than stage I patients in the post-lockdown period (stage III, SOR = 1.39, 95%CI 1.19–1.62; stage IV, SOR = 1.44, 95%CI 1.26–1.63) (Figure 4B,C). The analysis of stage III melanomas showed a moderate but not significant level of heterogeneity (I^2^ = 57.92%, *p* = 0.08) and a significative publication bias (Egger’s test *p* = 0.002). The adjusted SOR for publication bias remained significant (SOR = 1.21, 95% CI 1.01–1.46).

#### 3.2.4. Histopathological Subtype

A total of 16 studies reported the histopathological subtype of melanomas, including 125,884 patients, of which 77,718 were diagnosed in the pre-COVID-19 period and 30,327 in the post-lockdown period. The SOR of NM diagnosis was higher than the SSM in the post-lockdown phase (SOR = 1.19, 95%CI 1.07–1.32). However, there was a moderate but not significant heterogeneity across the studies (I^2^ = 56.75%), with no evidence of publication bias (Egger’s test *p* = 0.10). Contrariwise, no differences in the diagnoses of other melanoma subtypes were identified between the pre-COVID19 and post-lockdown periods (ALM vs. SSM: SOR = 1.08, 95%CI 0.97–1.20, I^2^ = 0%; LMM vs. SSM: SOR = 1.03, 95%CI 0.85–1.27, I^2^ = 49.6%) (Appendix A).

#### 3.2.5. Mitoses

Overall, six studies compared the presence of mitoses in pre-COVID-19 vs. post-lockdown melanomas, analyzing 5032 and 4593 patients, respectively (Appendix A). We calculated an SOR of 1.57 (95%CI 1.17–2.11), with high between-study heterogeneity (I^2^ = 68.78%;). By removing the study of Klepfisch et al. (2023) [44], the presence of mitoses was found to be significantly higher in melanomas diagnosed in the post-lockdown period (SOR = 1.82, 95%CI 1.46–2.28), with a relevant reduction in the heterogeneity (I^2^ = 29.35%).

## 4. Discussion

We conducted a systematic review that investigated the impact of the COVID-19 pandemic on possible diagnostic delays in cutaneous melanoma. Our meta-analysis found a worsening of all prognostic factors in melanomas diagnosed in the post-pandemic period. In detail, the summary ORs indicated a higher rate of thick, ulcerated, and high-stage new melanoma diagnoses in the post-lockdown phase than in the pre-COVID-19 period.

Since 2020, COVID-19 has affected global healthcare delivery, with a considerable impact on oncology, as many malignancies typically detected by routine screening have been missed. Marson et al. (2021) showed a 43% decrease in melanoma diagnosis during the COVID pandemic and estimated that 19,600 melanomas would be delayed in their initial diagnosis in the United States [63]. A reduction of at least 30% of diagnoses was registered in most European countries [21,23,24]. However, all studies noted a complete recovery of the rate of melanoma diagnosis in the post-lockdown period, confirming that there was no change in melanoma incidence in any period, only a delay in diagnosis.

It is known that diagnostic delays result in a more aggressive and invasive disease, characterized by worse prognostic factors, such as a high Breslow depth, ulceration, a high mitotic rate, and metastasis extent [64,65,66,67]. Tejera-Vaquerizo and Nagore simulated the course of melanoma growth through a predictive model based on melanoma thickness and estimated a proportion of 45% of tumors that would be upstaged with a 3-month delay, with a potential reduction in 10-year survival rate from 90% to 87.6% [68]. Our analysis showed a 31% higher rate of invasive melanoma diagnoses in the post-lockdown phase compared to in situ lesions, with no differences across the years or populations. Focusing on invasive melanomas, we found a 14% and 62% higher risk of melanoma with Breslow 1–2 mm and >2 mm, respectively, in the post-Lockdown period. Moreover, our meta-analysis showed a 35% increased risk for ulceration and 57% of mitoses at melanoma diagnosis were found in the post-lockdown period compared to the pre-COVID-19 period. Overall, we found a significant increase in the diagnosis of stage II, III, and IV tumors compared to stage I, with the risk increasing up to 44% for stage IV melanomas. Our results are in line with two recent meta-analyses [56,57]. Seretis and colleagues (2022) compared Breslow thickness, ulceration rate, and stage of melanomas diagnosed during the pre-COVID-19, lockdown, and post-lockdown periods in European countries, extracting data from 25 articles, with a total of 32,231 patients [56]. They found a significant increase in thickness, ulceration rate, and stage III melanoma in the post-lockdown group, as in our study. Recently, a meta-analysis including 102,263 patients from 27 studies demonstrated a significant increase in ulceration rates but not in thickness in the post-pandemic period [57]. Differences in sample size and the categorization of variables could explain the discrepancies with our results. Indeed, the authors assessed the long-term effects of the pandemic, thus excluding studies describing the period March–July 2020. They calculated the cumulative estimates for Breslow thickness and ulceration from 9 and 8 studies, respectively, not considering the data from research letters, case reports, and case series articles. Instead, based on our selection criteria, we analyzed thickness as a categorical variable, and also used Breslow depth and staging data to extract information on in situ or invasive melanoma diagnoses, and estimated the SOR for ulceration from 24 studies.

Overall, the results of our meta-analysis support a COVID-related diagnostic delay, as we demonstrated a consistently higher risk of all the worst prognostic factors in melanoma diagnosed after the COVID-19 pandemic. The disruption of healthcare services led to a greater proportion of melanomas being diagnosed with delay compared to the pre-pandemic period, as was also observed for cancers at other sites, including breast, lung, bladder, and colon carcinomas [69]. Consequently, we could expect to observe an increase in melanoma mortality in the coming years. Alternatively, we may speculate that diagnostic activities were redirected towards prioritizing more suspicious cases during the pandemic, while diagnostic pressure on less suspicious cases slightly decreased. If this alternative explanation is correct, an increase in melanoma mortality may not occur or turn out to be milder than expected, especially considering that overdiagnosis is possible and frequent for melanoma. Therefore, while our findings showing a worsening of all prognostic factors are extremely worrying, only the monitoring of temporal trends (e.g., in cancer registries) will reveal the real impact of the COVID-19 pandemic on melanoma mortality.

The argument that diagnostic delays during the pandemic disproportionately affected more severe cancer cases is supported by several studies that emphasize how disruptions in healthcare systems led to delays in the detection and treatment of more advanced-stage cancers. This is likely because early-stage cancers are often asymptomatic or present with less urgent symptoms, which may have led patients to postpone care, while more severe cases eventually required medical attention due to their more aggressive symptoms. Vanni et al. (2021) found that delayed diagnoses during the pandemic led to patients presenting with larger tumors and more node-positive breast cancers, indicating that more severe cases became prevalent [70]. The study reported an increase in average tumor size at diagnosis and a higher incidence of cancers with lymph node involvement. This trend suggests that patients with advanced disease, who are more likely to exhibit symptoms, were eventually diagnosed, whereas patients with early-stage cancers may have been missed due to the delays. Rutter et al. (2021) found that during the pandemic, there was a rise in emergency presentations of cancer, where patients were diagnosed after coming to the hospital for severe symptoms rather than through screening or routine visits [71]. These cases are often more advanced because emergency presentations typically occur when the disease has progressed to a symptomatic stage. The study reported that during the peak of the pandemic, the proportion of colorectal cancer diagnoses made through emergency routes rose by approximately 11%, reflecting a shift toward diagnosing more severe, symptomatic cases that required urgent medical attention.

The consistency of the results in our meta-analysis must be considered in light of some important limitations. First, the geographical distribution of the studies is somewhat skewed, with half of the studies coming from three countries (Italy, USA, and Ireland) which, although inhabited by populations of predominantly European origin, are not those with the highest melanoma incidence and mortality, which occurs in Australia, New Zealand and Northern European countries. Therefore, the picture that we obtained may not be representative of what occurred globally during the pandemic. Moreover, we should consider the moderate-to-substantial heterogeneity that affects most meta-analysis estimates. The interpretation of this is challenging. It may result from the different designs employed across the published studies but may also reflect differences in the severity of lockdown and/or in the efficacy with which diagnostic procedures were reorganized during the COVID-19 pandemic (how resilient the system was, how well the crisis was managed, etc.). Other sources of diversity across studies may be relevant, including the variability in the start and end date of the periods that were labeled pre- and post-pandemic, the failure to report on outcomes other than tumor thickness in many of the included studies, and the heterogeneity in the age distribution of melanoma patients, which might suggest different clinical settings.

While the early phase of the COVID-19 pandemic has been extensively studied, data regarding the late pandemic (from mid-2021 onwards) are still somewhat limited, though important trends are emerging. Studies during the late pandemic phase aim to understand the long-term effects of initial delays in diagnosis and treatment, as well as the recovery of healthcare services. Many countries have experienced a partial recovery in cancer screenings and diagnostic services, but a backlog of missed appointments and delayed diagnoses from the early pandemic period has continued to affect patient outcomes. Studies have indicated that while screening programs have resumed, they have struggled to address the delayed cases from the early pandemic. This backlog means that a significant number of cancers that should have been diagnosed earlier are only being detected now, often at more advanced stages [72].

Although many studies have not yet provided comprehensive data on long-term progression-free survival (PFS) and overall survival (OS) from late pandemic diagnoses, early findings suggest that the initial delays are likely to contribute to ongoing reductions in survival. For instance, cancers diagnosed at advanced stages during the pandemic continue to show poorer survival rates as treatment options become more limited for advanced disease. In particular, there are two key aspects to be considered. On one hand, advanced-stage cancers represent a higher percentage of the total due to delayed diagnoses, leading to a worsening of overall survival (OS), even when not stage-specific. On the other hand, stage-specific OS for melanomas also deteriorated, as patients experienced delayed or reduced access to therapy and follow-up care activities. As these delayed cases are now being treated, oncologists are starting to observe the negative impact on survival outcomes. However, these findings are still emerging, and full survival data (5-year or 10-year OS) will take more time to become available [73].

The late pandemic phase also saw variability in patients’ access to cancer treatment. Some countries were able to stabilize their healthcare systems more effectively, while others continued to experience disruptions due to new COVID-19 variants and healthcare worker shortages. A growing body of evidence suggests that for patients diagnosed during the late pandemic, delays in treatment were less common than for those diagnosed earlier in the pandemic, but uneven access to treatment based on geographic region or healthcare resource availability has continued to affect outcomes, especially for underserved populations. This variability in care means that some regions are seeing ongoing diagnostic delays and reduced survival rates, while others are beginning to recover and return to pre-pandemic diagnostic patterns [74]. In conclusion, we found that the disruption to healthcare systems caused by the COVID-19 pandemic resulted in a significant shift in diagnosis towards more advanced lesions with a less favorable prognosis. Monitoring cause-specific mortality in the coming years will allow us to more accurately quantify the real impact of the pandemic on the burden of melanoma in different countries around the world. In any case, health systems must learn from these lessons and take steps to strengthen their resilience capacity, for example, by creating and regularly updating pandemic preparedness plans to limit the impact of any future event on cancer care.

## Figures and Tables

**Figure 1 cancers-16-03734-f001:**
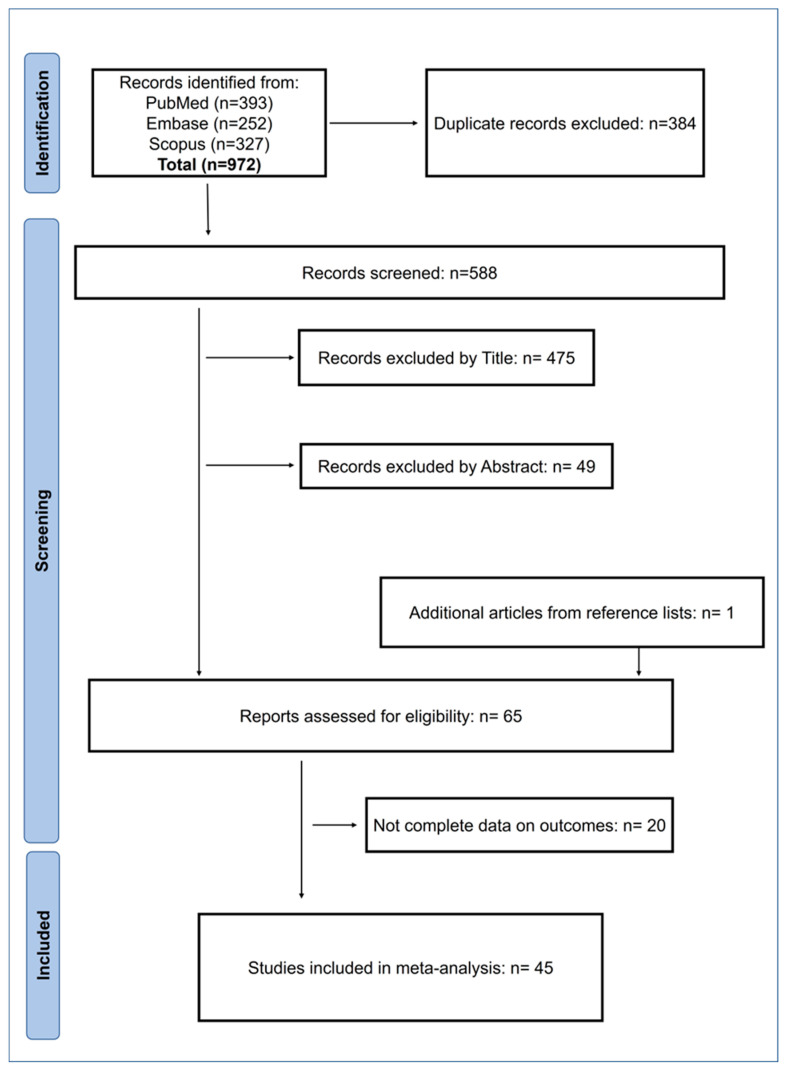
Flow-chart of the selection process for the studies included in the literature review and meta-analysis.

**Figure 2 cancers-16-03734-f002:**
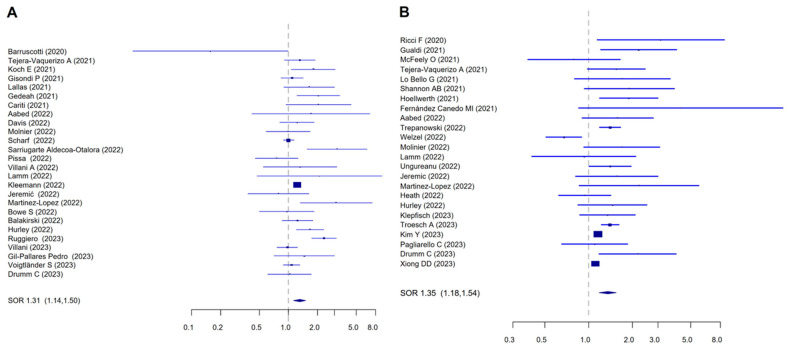
Forest plot for the association between the occurrence of (**A**) invasive melanoma (vs. in situ, taken as reference) and (**B**) ulceration (vs. the absence of ulceration) with the diagnosis of melanoma in the post-lockdown period. (**A**) I^2^ = 74.45, Begg *p* = 0.38, Egger *p* = 0.47; (**B**) I^2^ = 78.39, Begg *p* = 0.29, Egger *p* = 0.014.

**Figure 3 cancers-16-03734-f003:**
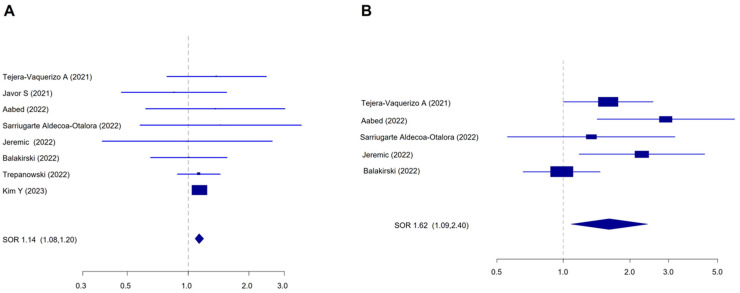
Forest plot for the association of high Breslow thickness with the diagnosis of melanoma in the post-lockdown period. (**A**) Breslow 1–2 mm vs. Breslow < 1 mm, I^2^ = 0%, Begg *p* = 1.0, Egger *p* = 0.95; (**B**) Breslow > 2 mm vs. Breslow < 1 mm I^2^ = 56.3%, Begg *p* = 0.48, Egger *p* = 0.18.

**Figure 4 cancers-16-03734-f004:**
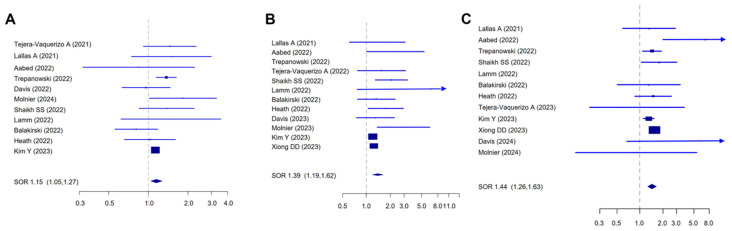
Forest plot for the association of melanoma AJCC stage and the diagnosis in the post-lockdown period. (**A**) Stage II vs. Stage I, I^2^:11.94%, Begg *p* = 0.87, Egger *p* = 0.92; (**B**) Stage III and vs. Stage I, I^2^:57.92%, Begg *p* = 0.04, Egger *p* = 0.002; (**C**) Stage IV vs. stage I, I^2^:4.81%, Begg *p* = 1.0, Egger *p* = 0.17.

**Table 1 cancers-16-03734-t001:** Characteristics of the studies included in the systematic review and meta-analysis.

Paper ID	First Author (Reference)	Publication Year	Country	Dates of Study Periods	Participants	Age, Years (SD)	Sex		Reported Outcomes *
							*M*	*F*	
1	Barruscotti [8]	2020	Italy	February 2019–May 2019	42	63 (16)	23	22	Breslow thickness
				February 2020–May 2020	6	63 (16)	5	1	
2	Ricci [9]	2020	Italy	January 2020–March 2020	158	NR	NR	NR	Histological subtype and ulceration
				March 2020–May 2020	34				
				May 2020–June 020	45				
				January 2021–June 2021	294				
3	Cariti [10]	2021	Italy	May 2017–June 2017	51	61.0	31	20	Breslow thickness Histological subtype
				May 2018–June 2018	41	62.0	20	21	
				May 2019–June 2019	48	61.0	31	17	
				May 2020–June 2020	32	55.0	16	16	
4	Fernández-Canedo [11]	2021	Spain	April 2019–August 2019	48	NR	NR	NR	Ulceration Mitosis
				April 2020–August 2020	18				
5	Gedeah [12]	2021	Belgium	March 2018–December 2018	169	NR	NR	NR	Breslow thickness
				March 2019–December 2019	161				
				March 2020–December 2020	140				
6	Gisondi [13]	2021	Italy	March 2019–October 2019	634	61.0 (3.6)	351	283	Breslow thickness
				March 2020–October 2020	556	62.2 (3.6)	314	242	
7	Gualdi [14]	2021	Italy	March 2017–July 2019	220	NR	262	271	Ulceration Mitosis
				March 2020–July 2020	168				
8	Hoellwerth [15]	2021	Austria	March 2018–June 2018	428	61.0	228	200	Ulceration
				March 2019–June 2019	505	60.0	260	245	
				March 2020–June 2020	432	63.0	233	199	
9	Javor [16]	2021	Italy	January 2019–December 2019	138	NR	NR	NR	Breslow thickness
				January 2020–December 2020	87				
10	Koch [17]	2021	Greece	January 2019–March 2020	191	52.8	80 54	110 51	Breslow thickness
				April 2020–March 2021	105	53.3			
11	Lo Bello [18]	2021	Italy	March 2019–December 2019	104	NR	NR	NR	Ulceration
				March 2020–December 2020	91				
12	Lallas [19]	2021	Greece	2016–2019	165	58.7 (15.1)	140	130	Breslow thickness Tumor staging
				2020	105	51.1 (11.4)			
13	McFeely [20]	2021	Ireland	2019	78	68.5 75.5	73	89	Ulceration
				2020	84				
14	Shannon [21]	2021	USA	June 2019–August 2019	172	68.0	96	76	Ulceration Mitosis
				June 2020–August 2020	153	68.0	88	65	
15	Tejera-Vaquerizo [22]	2021	Spain	March 2019–June 2019	303	64.0 (16.4) 62.9 (16.7)	NR	NR	Breslow Thickness Ulceration Tumor staging
				March 2020–June 2020	164				
16	Aabed [23]	2022	Romania	January 2018–December 2019	163	58.1 (16.3) 58.8 (15.9)	157	144	Breslow thickness Ulceration Tumor staging Histological subtype
				January 2020–December 2020	138				
17	Balakirski [24]	2022	Germany	January 2019–December 2019	320	63.7 (17.7)	NR	NR	Breslow thickness Tumor staging
				January 2020–December 2020	319	63.0 (19.4)			
				January 2021–December 2021	347	65.7 (16.4)			
18	Bowe [25]	2022	Ireland	January 2019–December 2019	52	NR	73	90	Breslow thickness
				January 2020–December 2020	61				
				January 2021–December 2021	51				
19	Davis [26]	2022	USA	August 2019–March 2020	375	65.7	236	139	Breslow Tickness Tumor staging
				May 2020–December 2020	313	67.0	182	131	
				May 2019–June 2019	72	NR	NR	NR	
				January 2020–February 2020	101	NR	NR	NR	
				May 2020–June 2020	20	NR	NR	NR	
20	Gil-Pallares [27]	2022	Spain	March 2019–September 2019	29	59 (18)	14	15	Breslow thickness Tumor staging
				September 2019–March 2020	36	70 (17)	11	25	
				March 2020–September 2020	24	66 (17)	11	13	
				September 2020–March 2021	30	69 (16)	11	19	
21	Heath [28]	2022	UK	November 2018–March 2020	276	NR	135	141	Tumor staging Ulceration
				March 2020–March 2021	242		118	124	
22	Hurley [29]	2022	Ireland	March 2019–December 2019	277	68.5	137	140	Breslow thickness Ulceration
				March 2020–December 2020	312	63.1	146	166	
23	Jeremić [30]	2022	Serbia	January 2017–March 2020	311	64.5 (15.8)	174	137	Breslow thickness Ulceration Histological subtype
				March 2020–March 2022	82	65.7 (15.3)	46	36	
24	Kleemann [31]	2022	Germany	March 2019–March 2020	35,037	NR	19,305	15,732	Breslow thickness
				March 2020–March 2021	32,189		17,414	14,775	
25	Lamm [32]	2022	USA	May 2019–May 2020	51	61.3 (2.1)	32	19	Breslow thickness Ulceration Histological subtype Tumor staging
				June 2020–September 2021	61	63 (2)	32	29	
26	Martinez-Lopez [33]	2022	Spain	March 2019–March 2020	77	63.3 (1.9)	43	34	Breslow thickness Ulceration
				March 2020–March 2021	53	65.0 (2.3)	23	30	
27	Molinier [34]	2022	France	March 2019–October 2019	257	NR	NR	NR	Breslow thickness Ulceration Tumor staging Histological subtype Mitosis
				March 2020–May 2020	55				
				May 2020–October 2020	181				
28	Pissa [35]	2022	Sweden	April 2019–March 2020	126	NR	NR	NR	Breslow thickness
				April 2020–March 2021	118				
29	Sarriugarte Aldecoa-Otalora [36]	2022	Spain	March 2018–October 2019	155	NR	NR	NR	Breslow thickness
				March 2020–October 2020	55				
30	Scharf [37]	2022	Europe	2019–2020	2311	NR	NR	NR	Breslow thickness
				2020–2021	1722				
31	Shaikh [38]	2022	USA	March 2019–March 2020	246	NR	130 137	116 109	Tumor staging Histological subtype
				March 2020–January 2021	246				
32	Trepanowski [39]	2022	USA	January 2019–February 2020	2062	65.5	1236 1053	826 781	Breslow thickness Ulceration Tumor staging Histological subtype
				January 2020–February 2021	1834	65.8			
33	Ungureanu [40]	2022	Romania	March 2019–February 2020	341	59.0	164	177	Ulceration Histological subtype
				March 2020–February 2021	275	63.0	138	137	
34	Villani [41]	2022	Italy	2018	216	55.4	13	17	Breslow thickness
				2019	294	59.2	21	23	
				2020	233	55.9	27	33	
				2021	288	57.3	22	25	
35	Welzel [42]	2022	Germany	January 2019	327	NR	NR	NR	Ulceration
				January 2020	319				
				January 2021	295				
36	Drumm [43]	2023	Ireland	January 2019–December 2019	117	NR	58	59	Breslow thickness Ulceration Histological subtype
				January 2020–December 2020	117		59	58	
				January 2021–December 2021	110		57	53	
37	Klepfisch [44]	2023	France	January 2019–March 2020	490	64.1 (14.5)	267	223	Ulceration Histological subtype Mitosis
				March 2020–May 2020	60	62.2 (16.2)	38	22	
				May 2020–November 2020	496	63.1 (15.1)	269	227	
38	Demaerel [45]	2023	Belgium	2017	1029	NR	NR	NR	Histological subtype
				2018	1007				
				2019	884				
				2020	961				
				2021	1218				
39	Kim [46]	2023	USA	2019	20,074	65 (12)	11,844	8230	Breslow thickness Ulceration Tumor staging Histological subtype
				2020	16,945	65 (12)	9828	7117	
40	Pagliarello [47]	2023	Italy	March 2019–March 2020	402	59.4 (17.3)	213	189	Ulceration
				March 2020–March 2021	246	60.8 (17.7)	117	129	
41	Ruggiero [48]	2023	Italy	March 2019–March 2020	298	NR	NR	NR	Breslow thickness
				March 2021–March 2022	560				
42	Troesch [49]	2023	Europe	September 2018–March 2020	4340	62.3 (16.2)	2300	2040	Ulceration Histological subtype Mitosis
				March 2020–March 2020	3525	63.2 (15.6)	1868	1657	
43	Villani [50]	2023	Italy	January 2018–December 2018	216	55.4	94	122	Breslow thickness
				January 2019–December 2019	294	59.2	140	154	
				January 2020–December 2020	233	55.9	105	128	
				January 2021–December 2021	288	57.3	135	153	
44	Voigtländer [51]	2023	Germany	March 2019–February 2020	1392	NR	NR	NR	Breslow thickness
				March 2020–February 2021	1158				
45	Xiong [52]	2023	USA	2018–2019	42,034	NR	NR	NR	Ulceration Tumor staging Histological subtype
				2020	17,984				

* The column contains only the reported outcomes that were used in our analysis. NR, not reported; SD, standard deviation.

## Data Availability

Data are available upon request to the corresponding author.

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
