# Peer review of "Impact of COVID-19 Pandemic on Delay of Melanoma Diagnosis: A Systematic Review and Meta-Analysis"

_cancers, 2024, doi:10.3390/cancers16223734_

Round 1
Reviewer 1 Report
Comments and Suggestions for Authors
The authors have provided a manuscript titled "Impact of COVID-19 pandemic on delay of melanoma diagnosis." Below are some suggestions for consideration:
-The introduction section could further elaborate on the potential consequences of delayed melanoma diagnosis, such as the impact on mortality rates.
-The author utilized the Newcastle-Ottawa Scale (NOS) for assessment. A more detailed explanation of the evaluation criteria would help readers from other fields better understand the content of the paper.
-The author proposed an interesting point that diagnostic delays during the pandemic primarily affected more severe cases. Providing additional data to support this argument would be beneficial.
Comments on the Quality of English LanguageThe manuscript still requires some language polishing.
Author Response
The responses are below, thank you.

Reviewer 2 Report
Comments and Suggestions for Authors
This meta-analysis assessed the impact of COVID-19 on the diagnosis of melanoma. The authors showed that COVID-19 had negatively impacted the outcome of melanoma. Overall, this study is well-designed and the manuscript is well-written. Thus, I just have one minor comment.
1. Because most of inlcuded studies assessed the status of early pandemic, please add some discussion about the limited findings in the late pandemic.
Author Response
The responses are below, thank you.

Reviewer 3 Report
Comments and Suggestions for Authors
First of all, is this manuscript a systematic review or research article? Since it has materials & Methods and results.
1) I strongly recommend the authors draw an appropriate graphical abstract at the end of the introduction.
2) The authors should write how anti-covid-19 could cause delay in melanoma diagnosis (introduction,3rd paragraph)
3) Sections and sub-sections should be revised. E.g. What do outcomes mean on page 5?
4) I still do not find any novelty or reasonable achievement in this work since it just searched melanoma and COVID-19 from paper.
Author Response
The responses are below, thank you.

Round 2
Reviewer 3 Report
Comments and Suggestions for Authors
The manuscript has been improved. However, the authors did not provide a graphical abstract.